# In Situ Gel Containing *Lippia sidoides* Cham. Essential Oil for Microbial Control in the Oral Cavity

**DOI:** 10.3390/microorganisms13112585

**Published:** 2025-11-13

**Authors:** Maria Vitoria Oliveira Dantas, Quemuel Pereira da Silva, Alexandre Almeida Júnior, João Vitor Souto Araújo Queiroz, José Filipe Bacalhau Rodrigues, Rosana Araújo Rosendo, Marcus Vinicius Lia Fook, Paulo Rogério Ferreti Bonan, Francisco Humberto Xavier Júnior, Fábio Correia Sampaio

**Affiliations:** 1Graduate Program in Dentistry, Federal University of Paraíba, João Pessoa 58051-900, PB, Brazil; mvod@academico.ufpb.br (M.V.O.D.); quemuelpereira7@gmail.com (Q.P.d.S.); 2Department of Clinical and Social Dentistry, Federal University of Paraíba, João Pessoa 58051-900, PB, Brazil; alexandrejunior.02@hotmail.com (A.A.J.); paulo.bonan@academico.ufpb.br (P.R.F.B.); 3Materials Science and Engineering Department, Federal University of Campina Grande, Campina Grande 58429-000, PB, Brazil; joaovitorsoutoaraujoqueiroz@gmail.com (J.V.S.A.Q.); filipe.rodrigues@certbio.ufcg.edu.br (J.F.B.R.); marcus.liafook@certbio.ufcg.edu.br (M.V.L.F.); 4Academic Unit of Biological Sciences, Federal University of Campina Grande, Patos 58708-110, PB, Brazil; cesprodonto@hotmail.com; 5Laboratory of Pharmaceutical Biotechnology (BioTecFarm), Department of Pharmacy, Federal University of Paraíba, João Pessoa 58051-900, PB, Brazil; fhxj@academico.ufpb.br

**Keywords:** volatile oil, surgical wound infection, poloxamer, chitosan, microbial sensitivity tests, thermosensitive gel, oral mucoadhesion

## Abstract

Surgical site infections in oral and maxillofacial interventions are often exacerbated by biofilm formation, and current antimicrobial treatments are hampered by issues such as resistance and adverse effects. This article aimed to develop, characterize, and evaluate the antimicrobial activity of *Lippia sidoides* Cham. essential oil (LSEO) gel composed of poloxamer (P) and chitosan (C). Gas chromatography–mass spectrometry (GC-MS) analysis identified thymol as the major component of LSEO (71.04%). In situ P-gels containing LSEO (0.25–1.0%) were produced with and without C. The addition of C resulted in gels with nanometric particle sizes (263.8 ± 231 nm; PDI 0.39 ± 0.17) and a positive zeta potential (+4.81 ± 1.97 a + 8.19 ± 0.51 mV), exhibiting pseudoplastic behavior in rheological analysis. The sol–gel transition temperature (Tsol–gel) was found to be between 20 and 28 °C, with a transition time at 37 °C ranging from 18.76 ± 1.24 s to 46.46 ± 8.89 s. LSEO showed MIC values of 256, 128, and 128 µg/mL against *Staphylococcus aureus*, *Escherichia coli*, and *Candida albicans*, respectively, while in situ LSEO gels presented MIC values above 5 µg/mL for all tested strains. Therefore, the developed gel containing LSEO showed promising application in dentistry, offering a potential new treatment perspective for surgical site infections in oral and maxillofacial surgery.

## 1. Introduction

Microbial biofilm accumulation is a major cause of surgical site infections (SSIs) in oral and maxillofacial surgeries. The oral cavity is a semi-contaminated environment, and the mechanical control of biofilm is hampered by postoperative pain and discomfort [1]. To prevent SSIs, antimicrobial agents like triclosan-coated sutures and chlorhexidine-based antiseptics have been used. However, triclosan’s therapeutic use is limited by concerns over bacterial resistance from widespread use. Chlorhexidine, the gold-standard antimicrobial for post-surgical care, is effective in reducing microbial load, but it can cause adverse effects with prolonged use [2,3].

Essential oils (EOs) are promising antimicrobial agents that help combat microbial resistance. Their hydrophobic nature and diverse compounds reduce resistance risks due to multiple mechanisms of action [4]. Therefore, their therapeutic effects are applied to oral infections, such as caries, candidiasis, gingivitis, and periodontitis, as an alternative to antibiotics [5,6].

Studies have focused on encapsulating the essential oil of *Lippia sidoides* Cham. “pepper rosemary”, a shrub from Northeastern Brazil. This oil, rich in thymol and carvacrol, exhibits antibacterial, antifungal, antioxidant, anesthetic, and anti-inflammatory properties. Its antimicrobial activity has been demonstrated against *S. aureus*, *E. coli*, and *C. albicans* [7,8,9].

The oral mucosa is attractive for the delivery of therapeutic agents to its accessibility, rich blood supply, bypass of first-pass metabolism, rapid healing, and permeability. However, conventional therapies like antimicrobial rinses are quickly removed by saliva [10,11]. In situ gels, which transition from solution to gel upon application, offer advantages. Mucoadhesive gels, in particular, allow for spray application, prolong mucosal contact time, reduce dosing frequency, and improve treatment adherence and patient comfort [10,11].

Poloxamers are thermosensitive polymers widely used in pharmaceutical formulations, including in situ gels. These ABA-type triblock copolymers consist of polyoxyethylene (A) and polyoxypropylene (B) units. Poloxamer 407 (P) is a non-ionic, water-soluble copolymer that gels at body temperature (37 °C) when used at 15–30%. Its amphiphilic nature, with water-soluble (A) units and liposoluble (B) units, provides surfactant properties [12]. In addition, P offers good solubilization, low toxicity, non-irritation to biomembranes, effective drug release, and compatibility with various biomolecules and excipients [13,14].

Poloxamer gels often dissolve quickly in the oral mucosa, limiting their duration. To address this, poloxamer 407 (P) is combined with mucoadhesive polymers like chitosan (Q), which interacts with P’s polyoxyethylene units to enhance gel resistance, mucoadhesion, and residence time [15,16,17]. This study aims to encapsulate *Lippia sidoides* essential oil (LSEO) in a P-Q gel to achieve thermogelling, stability, and improved mucoadhesion, thus protecting the oil’s volatile components and boosting its antimicrobial activity. The goal is to develop, characterize, and assess the antimicrobial efficacy of in situ LSEO-loaded PC gels for controlled oral mucosal release.

## 2. Materials and Methods

### 2.1. Materials

Poloxamer 407 (purity > 99%, molar mass 12,600 g/mol, 30% PPO), low-molecular-weight chitosan (>75% deacetylation, molecular weight 50–190 KDa), sodium tripolyphosphate (TPP) (technical grade 85%, molecular weight 367.86 g/mol), Tween 80, chlorhexidine, resazurin, and nystatin were acquired from Sigma-Aldrich (Saint Louis, MO, USA).

### 2.2. Lippia Sidoides Cham. Essential Oil Extraction

The adult shrubs of *Lippia sidoides* Cham. were collected from the Medicinal Plants Garden of UFPB at the Institute of Research in Pharmaceuticals and Medicines (IPeFarM), João Pessoa, Paraíba, Northeastern Brazil (latitude, 7°6′55″ south; longitude, 34°51′40″ west) in March 2019 between 06:00 and 07:00 h. The botanical identification of the plant was carried out, and the exsiccate was incorporated under the number JPB 47237 into the collection of the Prof. Lauro Pires Xavier Herbarium at UFPB, with SisGen registration number ABDE352, in compliance with the provisions of Law No. 13.123/2015 and its regulations. The technique used to extract the oil was steam extraction of the aerial parts of the plant using a mini essential oil distiller (Model D2—v5.2-Linax). The density of the LSEO was determined using a pycnometer at 25 °C in triplicate, and the pH was checked by potentiometry in four repetitions.

### 2.3. Gas Chromatography Coupled with Mass Spectroscopy (GC-MS)

The chemical composition of the essential oil was determined by gas chromatography coupled with mass spectroscopy (GC-MS), carried out on a Shimadzu model GCMS-QP 2010 Ultra (Shimadzu Corporation, Kyoto, Japan) with an RTX-5MS capillary column (5% diphenyl/95% dimethylpolysiloxane) with a size of 30 m (length)/0.25 mm internal diameter/0.25 µm. The sample injection volume was set to 1 µL, with a split ratio of 1:150. Helium was utilized as the carrier gas at a flow rate of 1 mL/min. The sample injection port temperature was set at 300 °C, and the injector temperature was maintained at 280 °C. The oven temperature was initially programmed to 100 °C, held for 5 min, then increased to 280 °C at a rate of 10 °C/min, and maintained at 280 °C for an additional 5 min. An electron impact ionization source was employed at 70 V, with a full scan mode over a mass range of 35–400 *m*/*z*. The constituents were identified by searching for data found in the literature and the equipment’s library (Databases: NIST2008|NIST2008 + Shimadzu|FFNSC 1.3).

### 2.4. Preparation of In Situ Gels

The development of in situ gels took place in steps [18] (Table 1). The first step consisted of preparing 20% P gels in ice-cold deionized water under magnetic stirring (300 rpm) (model HJ-5-220V–ION, Changzhou, China) for 10 min. The gels were subjected to 3 cycles of refrigeration/agitation.

These cycles were carried out because P407 has a sol–gel transition at temperatures close to the ambient temperature. As a result, gel formation was observed due to the prolonged stirring time and the gradual increase in the system’s temperature due to mechanical friction. In order to avoid early gelation and keep the polymer in a liquid state, which is necessary for the proper homogenization of the components, it was necessary to intersperse the stirring cycles with cooling steps, ensuring the stability and uniformity of the system during preparation.

After complete dispersion of the P powder, the gels were kept refrigerated at 6 °C without stirring for 24 h. The second step was the dispersion of the LSEO in the P gel. LSEO was solubilized in 5% Tween 80 at final concentrations of 0.25, 0.5, and 1%. The oil was added to the gels under magnetic stirring (600 rpm) for 10 min. The dispersions were then subjected to two cycles of refrigeration/magnetic agitation to obtain maximum interaction between the LSEO and P.

For the mucoadhesive formulation, a 1 mg/mL solution of chitosan was prepared in glacial acetic acid solution (1% *v*/*v*) under magnetic stirring (300 rpm) for 24 h. Posteriorly, 2.4 mL of this solution was poured into the dispersions corresponding to each group following magnetic stirring (300 rpm) for 20 min at room temperature [18]. Finally, 200 µL of sodium tripolyphosphate solution (TPP, 1.5% *m*/*v*) was poured into the dispersions. A total of 10 mL of each resulting gel was kept under constant magnetic stirring at 300 rpm for 60 min.

### 2.5. Characterization of the Gels

Fourier-Transform Infrared Spectroscopy (FTIR) analysis was used to evaluate possible interactions between poloxamer and chitosan and to verify the incorporation of LSEO into the gels. The spectra were acquired on a Perkin Elmer Spectrum 400 Spectrometer (Waltham, MA, USA) coupled with an Attenuated Total Reflectance (ATR) accessory equipped with a thallium bromo-iodide crystal (KRS-5, American Elements, Los Angeles, CA, USA). Analyses were performed in the range of 4000–400 cm^−1^ and consisted of 32 scans with a resolution of 4 cm^−1^. Before collecting the spectra, samples were dried for 24 h to remove the water present in the gels and, consequently, eliminate the interference of water bands.

The size and polydispersity index of the gels were analyzed by the dynamic light scattering method and the zeta potential (PZ), using the electrophoretic mobility technique associated with Henry’s equation by Zetasizer^®^ Lab (Malvern, Worcestershire, UK) equipped with a 633 nm laser beam at fixed optical scattering at 90° at 25 °C. Samples were diluted using ultrapure water (DLS) and 1 mM sodium chloride solution (PZ) in disposable cuvettes. The analyses were carried out in triplicate, and the results are expressed as mean ± standard deviation.

Scanning Electron Microscopy (SEM) analysis was performed after freeze-drying the samples. Thus, a Hitachi TM-1000 benchtop scanning electron microscope (Chiyoda-ku, Tokyo, Japan) was used with magnification up to 10,000-times, depth of focus of 1 mm, resolution of 30 nm, 15 KV low vacuum. and varying pressure (1 to 270 Pa), without metal coating.

The determination of the temperature and time of sol–gel transition (Tsol–gel) were carried out using the test-tube tilt method. A total of 2 mL of the gel (in solution phase) was inserted and kept in a water bath. To determine the temperature (Tsol–gel), the temperature of the water bath was slowly increased every 2 °C and was determined at the moment when the solution in the glass tube stopped flowing when subjected to inversion. For time (Tsol–gel), the temperature of the water bath was kept at 37 °C. Every 5 s, the test tube was removed and tilted to observe the state of the sample. The time required for gelation was determined by the lack of fluidity of the solution in the tube.

Rheological experiments are carried out to predict the viscoelastic and flow behavior of materials when subjected to mechanical stress. In the case of polymer gels intended for the medical field, they must have properties suitable for the mucosa for which they are intended [19]. Based on this principle, temperature scans were carried out from 26 to 40 °C, at a constant frequency of 1 Hz, to verify the state of gelation in the room–body temperature transition, and an oscillatory frequency scan from 0.1 to 10 Hz was performed to evaluate the apparent viscosity of the samples. The tests were carried out on a HAAKE^TM^ MARS^TM^ rheometer (Thermo Fisher Scientific, Waltham, MA, USA) equipped with a PP35Til rotor with plate-to-plate geometry (Waltham, MA, USA).

For the stability study, samples were initially subjected to a centrifugation test (Model 80-2b–MYLABOR, Quimis Aparatos Científicos Ltda., Diadema, SP, Brazil) at 3000 rpm (1300× *g*) for 30 min. Each gel (2 mL) was placed in Falcon tubes at room temperature and subjected to centrifugation. The purpose of the test is to stress the formulation, simulating an increase in the force of gravity and increasing the agitation of the particles. After the test, the product must remain stable, and any signs of instability must be observed. After centrifugation, formulations were subjected to stability studies at different storage conditions: 5 ± 2 °C, room temperature (25 ± 2 °C)*,* and 45 ± 2 °C for 30 days in sealed glass bottles. The physicochemical stability of the gels was observed by considering the pH, color, phase separation, and odor.

### 2.6. Microbiological Tests

The microorganisms tested were *Staphylococcus aureus* (ATCC 15656), *Escherichia coli* (ATCC 25922), and *Candida albicans* (ATCC 76485). The bacterial strains were grown in Brain Heart Infusion (BHI—Sigma-Aldrich, St. Louis, MO, USA), and the fungal strain was prepared in Sabouraud Dextrose Broth (CSD) (KASVI1, Kasv Imp e Dist de Prod/Laboratorios Ltda., Curitiba, Brazil), and both were incubated under aerobic conditions in a microbiological oven at 35 °C (fungi) and 37 °C (bacteria) for 24 h. The inoculums were standardized to 0.5 on the McFarland Scale.

### 2.7. Minimum Inhibitory Concentration (MIC)

The microdilution technique against bacterial strains was carried out according to the Clinical and Laboratory Standards Institute (CLSI, 2012) [20], with modifications, and the microdilution technique performed with the fungal strain was performed according to CLSI 2008, with modifications [21]. LSEO was solubilized in 5% Tween 80 and diluted with BHI or CSD to reach concentrations ranging from 2.048 to 16 µg/mL. Formulations with P and with PL704 + Q had LSEO in the range from 20 to 2.25 µg/mL diluted in the medium. Moreover, 0.9% saline solution, 5% Tween 80, and empty P and PC were used as negative controls, while 0.12% chlorhexidine and Nystatin (48 µg/mL) were used as positive controls. The plates were incubated in a microbiological oven for 24 h at 35–37 °C. After this period, MICs against bacterial strains were determined visually by oxidation–reduction of the indicator resazurin (0.1%). MIC against the fungal strain was read from the observation of cell aggregates at the bottom of the wells. The lowest concentration of LSEO at which there was no visible microbial growth was recorded as the MIC. The tests were carried out in triplicates.

## 3. Results and Discussion

### 3.1. Characterization of Lippia Sidoides Cham. Essential Oil (LSEO)

The LSEO extracted from the aerial parts of the plant had a density of 0.9432 g/mL and pH of 5.27. The chromatographic analysis resulted in the identification of thymol (71.04%), isopropyl p-cymene (10.22%), E-caryophyllene (6.32%), caryophyllene oxide (1.28%), and β-bisabolene (1.31%) as the main compounds (Table 2). Other compounds were also detected in lower concentrations, such as carvacrol (0.43%). Phenolic compounds commonly found in LSEO can vary according to natural factors. Thymol was expected to be the main compound given that previous studies have shown thymol to be the main component of LSEO from the northeast of Brazil [22,23].

### 3.2. Characterization of the Gels In Situ

The gels produced in this research and their respective compositions are described in Table 1.

P and PC gels appeared viscous, odorless, and translucent, and with a homogeneous structure at room temperature. The PLS0.2, PLS0.5, PLS1, PCLS0.2, PCLS0.5, and PCLS1 formulations had a viscous appearance, a characteristic odor (attributed to LSEO), a homogeneous structure at room temperature, and a slightly yellowish color, although the PLS1 group had the strongest yellowish color (attributed to the higher concentration of LSEO). The pH of the formulations without chitosan ranged from 6.7 to 6.9, while the formulations containing chitosan varied between 5.7 and 5.8.

The centrifugation test was carried out as a preliminary analysis to detect instability characteristics of the formulations, such as the separation of the components into phases. After the test, no visual changes were observed in the original appearance of the gels evaluated, suggesting stability.

Infrared spectra of the LSEO gels are presented in Figure 1. P, PLS0.2, PLS0.5, and PLS1 gels showed a similar spectral profile. All the samples showed vibration bands characteristic of poloxamer: vibration bands located between 2980 and 2880 cm^−1^ refer to the symmetrical and asymmetrical vibrations of –CH_3_ e –CH_2_ [7]; and the vibration band at 2880 cm^−1^ corresponds to the stretching vibration of the –CH_2_ group 2 [24]. The vibrations at 1467 and 1101 cm^−1^ are associated with the C–O– bonds belonging to the ether group [25], and the vibration at 961 cm^−1^ belongs to the C–O–C bond [8]; C–H stretching vibrations of alkanes are observed at 1342 cm^−1^ [7]. All these bands are characteristic of the molecular structure of poloxamer, which is a copolymer composed of a central hydrophobic polyoxypropylene chain flanked by two hydrophilic polyoxyethylene chains [7].

The spectra of PLS0.5, PLS0.2, and PLS1 showed new vibration bands after the incorporation of LSEO into the poloxamer gels (Figure 1b). The vibration band at 808 cm^−1^ is attributed to the torsional vibrations of the C–H bond (C–C–C–H); the vibration at 741 cm^−1^ is related to the combination of C–C stretching and C–C–C bending vibrations, both of which are present in aromatic rings [26]; vibrations located at 727 and 596 cm^−1^ are related to the C–C– bond in its torsion mode [9]. The vibration bands at 720, 649, and 640 cm^−1^ are characteristic of the vibrations of the other compounds present in the LSEO [9,27].

After the incorporation of the LSEO into the PC gels to produce a mucoadhesive system, it was possible to identify C and P interactions (Figure 1c,d). PC, PCLS1, PCLS0.2, and PCLS0.5 gels showed all bands characteristic of poloxamer (Figure 1c). Comparing the spectra of P and PC, new bands at 1577, 1413, 649, and 620 cm^−1^ correspond to chitosan. The chitosan amide II vibration band, commonly identified at 1590 cm^−1^ in the literature [28], was observed at 1577 cm^−1^, suggesting interactions between the amide group from chitosan and O–H bond from poloxamer [29]. The band at 1577 cm^−1^ is the amine II vibration of chitosan that is slightly shifted from 1590 to 1577 cm^−1^, indicating interactions between the amide group from chitosan and O–H bond from poloxamer [29]. The vibration at 1413 cm^−1^ is the symmetrical deformation of –CH_3_ in chitosan [28]. Furthermore, the vibration bands at 649 and 620 cm^−1^ correspond to C–H bending vibrations in aromatic rings. Finally, the vibration bands at 808, 741, 727, 720, and 596 cm^−1^ are associated with the LSEO (Figure 1d) [9,27]. Taken together, infrared results indicated interactions between chitosan and poloxamer, as well as confirming the incorporation of LSEO into the polymer matrix.

The particle size, polydispersity index, and zeta potential of each gel formulation were determined (Figure 2). The particle size of the samples increased with the presence of chitosan in the formulation, which also changed the polydispersity index (PDI). This was observed in the PLS0.5 group, which presented homogeneity (153.7 ± 1.7 nm; PDI 0.22 ± 0.03); however, when chitosan (PCLS0.5) was added, the sample became heterogeneous (263.8 ± 231; PDI 0.39 ± 0.17). The increase in the particle size can be associated with the higher molecular weight of chitosan. In addition, TPP forms cross-links that promote the formation of larger particles, leading to aggregation and reducing the homogeneity of the formulations [30,31].

The PLS0.2 and PCLS0.2 groups, formulated with the lowest concentration of EO tested in this study, presented a larger size (Figure 2a), but the increase in the amount of EO did not present a pattern of change in particle size; thus, it is suggested that the amount of EO did not interfere with the increase in particle size.

About PZ, EO interfered with the charge of particles, so that with the increase in the amount of EO, the positive charge of the PZ of the particles decreased. Gels without chitosan showed negative values (from −9.83 ± 0.55 to −5.65 ± 0.61 mV), while gels with chitosan showed positive values (from +4.81 ± 1.97 to +8.19 ± 0.51 mV). A positive charge of gels can be associated with the polycationic nature of chitosan. The mucoadhesive properties of chitosan are mainly based on the interaction between the positive charge of chitosan and the negative charge of the sialic acid of mucin, and as a result, the drug’s residence time on the mucosa is prolonged [32].

The micrographs obtained by Scanning Electron Microscopy (SEM) show the presence of an organized polymer matrix with lamellar structures (Figure 3). After the introduction of chitosan into the dispersion (Figure 3b), it is possible to notice the presence of lamellae with a denser morphology, with fewer cracks or fractures and a smooth surface. In the micrograph corresponding to the PCLS1 group (Figure 3c), it is possible to see the occurrence of refringence points, generated by the difference in the electronic density of the materials, indicating the presence of essential oil droplets dispersed in the polymer matrix.

The temperatures and gelling times of the formulations are shown in Table 3. In general, in situ gels should transition Tsol–gel close to body temperature (37 °C) in a short Tsol–gel time, remaining in solution form when stored at room temperature (25 °C) [33]. The formulations in this study had temperatures (Tsol–gel) ranging from 20 ± 1 °C to 28 ± 1 °C, with Tsol–gel time at 37 °C ranging from 18.76 ± 1.24 s to 46.46 ± 8.89 s. Usually. solutions with a P (15–30% w/v) are free-flowing liquids at 25 °C (room temperature) and are quickly converted to a gel state at 37 °C [34]. In addition, it can be seen that the temperature (Tsol–gel) decreased as the concentration of LSEO increased. Panomsuk et al. (2020) [35] also observed a decrease in Tsol–gel by increasing the concentration of *Syzygium aromaticum* essential oil in a P gel (20%).

Figure 4a displays the viscosity as a function of oscillatory frequency for the polymer gels. The curves indicate that all samples exhibit pseudoplastic behavior, characterized by high viscosity at low frequencies and a sharp decrease as oscillatory stress increases, followed by a tendency toward stability (second Newtonian plateau). This behavior is advantageous for medical applications, as the reduced viscosity under stress facilitates easier spreading during clinical administration [36,37].

Additionally, the sample PLSQ02, with the lowest concentration of essential oil, showed lower viscosity than the control (PC), unlike PLSQ05 and PCLS1, which had increased viscosity. These results suggest that at low concentrations, the oil acts as an external plasticizer, reducing viscosity by facilitating polymer chain flow. However, at higher concentrations, this effect reverses, likely due to the oil interfering with the gel matrix and hindering its organization and flow [38].

Figure 4b shows the tan δ graph, which represents the ratio between plastic loss and elastic storage moduli, as a function of temperature at constant frequency. The samples containing LSEO exhibit a tan δ below 1 across the analyzed temperature range, indicating a solid-like behavior without a Tsol–gel transition and, thus, demonstrating thermal stability and maintenance of the gel structure. In contrast, the PC sample shows a sol–gel transition around 32 °C, gelling at this point. Typically, aqueous poloxamer solutions exhibit a Tsol–gel transition around 37 °C, but this can shift due to other polymers or substances in the matrix. Here, the presence of chitosan lowered the transition temperature, and in combination with the essential oil, the behavior shift occurs outside the analyzed range, at lower temperatures, consistent with the literature [39,40,41].

After 7, 15, and 30 days in different storage conditions (25 ± 2 °C, 5 ± 2 °C, and 45 ± 2 °C), gels were evaluated by considering color, phase separation, odor, and pH (Table 4). Gels that retain their initial characteristics after production are considerably stable, with no phase separation or odor changes. PLS1, PLS0.5, and PLS0.2 stored at 45 ± 2 °C showed a darkening color. P + Q gels loaded with LSEO (PCLS1, PCLS0.5, and PCLS0.2) showed no changes in color, suggesting that chitosan may confer better stability to the formulations.

There was no significant change in the pH of the formulations over 30 days. The range of pH of the formulations was 5.64 to 7.15, which is close to the pH of commercial mouthwashes and falls within the accepted values of the European Standard for Oral Care Products (NEN-EN-ISO16408:2015) [42], which require mouthwashes to have a pH value between 3.0 and 10.5. Hence, there is minimal risk of oral irritation expected [43].

### 3.3. Antimicrobial Activity of the In Situ Gels

The results of the analysis of the antimicrobial activity of LSEO obtained by the microdilution method proved to be favorable in inhibiting the growth of the strains studied. For the *E. coli*, the MIC value found was 256 µg/mL. The MIC values showed the best results for Gram-positive *S. aureus* (128 µg/mL) and fungal strain *C. albicans* (128 µg/mL). The MIC values for the controls were 6 µg/mL for chlorhexidine and 3 µg/mL for nystatin.

Studies show that EOs inhibit microbial growth by affecting the phospholipid bilayer and the enzymatic mechanism of energy production and metabolism, damaging the structure and function of the bacterial cell membrane. In this way, the hydrophobic nature and chemical composition of EOs reduce the risk of microbial resistance, given the diversity of mechanisms of action that encompass a combination of compounds [4,44].

The MIC results of the present study corroborate Veras et al. (2017) [45]. In this study, the LSEO was extracted, identifying thymol (84.9%), ethyl-methyl-carvacrol (5.33%), and p-cymene (3.01%) as the main constituents. The MIC values found were 128 µg/mL for *S. aureus* and 512 µg/mL for *E. coli.* The mechanism of action of essential oils on Gram-positive bacteria and fungi seems to be similar. Brito et al. (2015) [46] evaluated the antifungal potential of LSEO using the microdilution technique and observed MICs ranging from 64 to 256 μg/mL against *Candida* strains. It is worth noting that the essential oil had thymol (71.04%) as its majority component. The greater sensitivity of Gram-positive organisms can be explained by the fact that Gram-positive bacteria have a single thick layer of peptidoglycan in their membrane, while Gram-negative bacteria have a more external membrane, acting as a greater molecular barrier [47].

The antimicrobial activity of the gels was assessed by the microdilution method (20 to 2.25 µg/mL of LSEO). P and PC formulations composed only of the polymers showed no antimicrobial activity, as expected. The P and PC gels containing LSEO inhibited microbial growth against the tested strains at MIC values as low as 5 µg/mL of LSEO.

Baldim et al. (2019) [48] encapsulated LSEO in a lipid nanosystem for antifungal activity. LSEO was found to be active against *C. albicans* (MIC = 156 µg/mL), while the nanosystem loaded with LSEO showed activity of 221 µg/mL. This effect was different from the one obtained in this study, where the incorporation of LSEO into gels provided the best antimicrobial action. Zanotto et al. (2023) [49] observed that mannosylerythritol lipid nanoemulsions enhanced the antimicrobial activity of essential oils. Among them, the essential oil of *Lippia sidoides* (OELS) showed MIC values of 500 µg/mL for *Escherichia coli*, 500 µg/mL for *Staphylococcus aureus*, and 125 µg/mL for *Candida albicans*.

In situ gel showed greater antimicrobial activity compared to the free LSEO with lower MIC values. This may be associated with the controlled release of the LSEO promoted by the polymeric system resulting in better bioavailability of the EO in the medium. In addition, the better inhibitory effects can be justified by the following possible reasons: (a) the LSEO microencapsulation process has the potential to act as a functional barrier, preventing physical and chemical reactions; (b) it allows the original characteristics of the LSEO to be maintained for a longer period when compared to free LSEO; and (c) the concentrations of antimicrobial substances (volatiles) remain constant for longer than other compounds present in essential oils [50,51].

Despite the need for further investigations into the mechanism of action, the interactions between gel particles and microorganisms, and the modulation of LSEO release, the MIC values obtained (128–256 µg/mL) indicate that the gels developed here are promising for the production of innovative commercial formulations containing LSEO for the treatment of SSIs.

Future studies should focus on completing the characterization of the gels, such as water absorption capacity and mucoadhesion, as well as evaluating the cytotoxicity of the LSEO-loaded gels to ensure safety and therapeutic potential. Additionally, investigating the release kinetics of LSEO will provide insights into its sustained action. These steps are essential for the clinical development of the formulation, paving the way for future clinical trials and potential therapeutic application.

## 4. Conclusions

The encapsulation of LSEO into in situ poloxamer + chitosan gel was successfully carried out and provided characteristics of potential mucoadhesion and stability, promoting the protection of the volatile constituents of the EO. Furthermore, the characterization methods confirmed the formulation of stable systems, with nanometer-scale particles that gel at temperatures from 20 °C. The gels enhanced the antimicrobial activity of LSEO against *S. aureus*, *E. coli*, and *C. albicans*. Thus, in situ gels can be easily applied to the oral mucosa as a spray to form a mucoadhesive gel film at body temperature. This form of application offers advantages such as convenient application, being able to reach difficult-to-access areas in the oral cavity, ensuring localized and controlled release of LSEO, and reducing pain and discomfort caused by surgical injuries, which highlights the high relevance and novelty of this study.

## Figures and Tables

**Figure 1 microorganisms-13-02585-f001:**
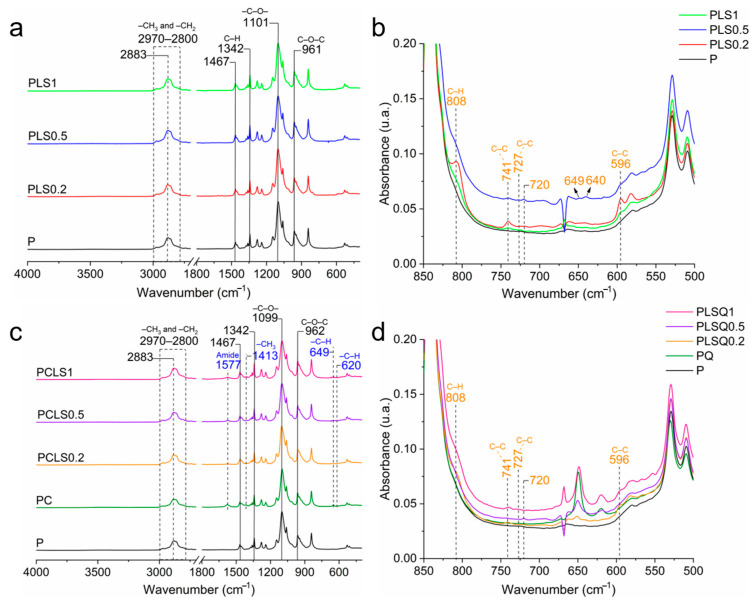
(**a**) Infrared spectra of the poloxamer gel with different concentrations of LSEO. (**b**) Detailed infrared spectrum of the gels at 850–500 cm^−1^. (**c**) Infrared spectra of the poloxamer/chitosan gel with different concentrations of LSEO. (**d**) Section of the infrared spectrum of the gels at 850–500 cm^−1^.

**Figure 2 microorganisms-13-02585-f002:**
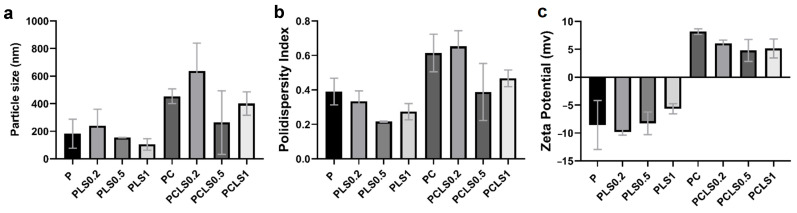
Results of the particle size (**a**), polydispersity index (**b**), and zeta potential (**c**) of each gel formulation.

**Figure 3 microorganisms-13-02585-f003:**
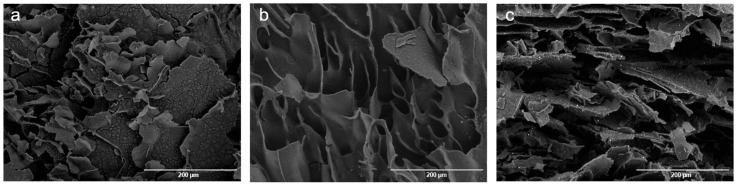
Scanning Electron Microscopy (SEM) micrographs: (**a**) P, (**b**) PC, and (**c**) PCLS1.

**Figure 4 microorganisms-13-02585-f004:**
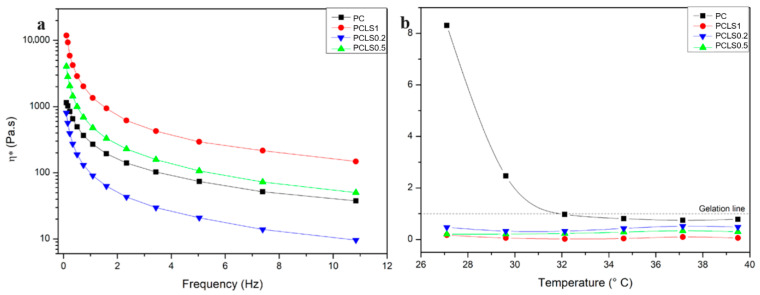
(**a**) Apparent viscosity as a function of the oscillatory frequency of the gels formed by poloxamer, chitosan, and LSEO in different proportions. (**b**) Tangent of loss (tan δ), ratio between plastic loss and storage moduli (G”/G’), as a function of temperature.

**Table 1 microorganisms-13-02585-t001:** Groups of in situ gels and final concentrations.

Groups	P407 (%)	TPP (%)	Q (%)	OELS (%)
P	20	0.03	-	-
PLS1	20	0.03	-	1
PLS0.5	20	0.03	-	0.5
PLS0.2	20	0.03	-	0.25
PC	20	0.03	0.024	-
PCLS1	20	0.03	0.024	1
PCLS0.5	20	0.03	0.024	0.5
PCLS0.2	20	0.03	0.024	0.25

Groups: P: P; PLS1: P + LSEO 1%; PLS0.5: P + LSEO 0.5%; PLS0.2: P + LSEO 0.25%; PC: P + C; PCLS1: P + C + LSEO 1%; PCLS0.5: P + C + LSEO 0.5%; PCLS0.2: P + C + LSEO 0.25%; (-) does not contain.

**Table 2 microorganisms-13-02585-t002:** Chemical composition of the OELS by GC-MS.

Compounds	Retention Time (min)	Concentration (%)	RI
α-Pinene	6.030	0.40	935
1-Octen-3-ol	6.877	0.37	982
β-Myrcene	7.070	1.77	993
4-Carene	7.535	0.66	1023
Isopropyl p-Cymene	7.678	10.22	1033
Limonene (l-isomer)	7.753	0.54	1038
Eucalyptol	7.805	0.34	1042
Isomeric geraniol	8.911	0.46	1119
Ipsdienol	9.501	0.50	1160
Terpinen-4-ol	9.913	1.31	1188
Methoxycymene	10.493	0.76	1314
Thymol	11.063	71.04	1472
Carvacrol	11.150	0.43	1496
β-Caryophyllene (E-caryophyllene)	12.218	6.32	1677
α-Bergamotene (trans)	12.270	0.41	1673
Guaiene derivative	12.368	0.90	1666
cis-α-Bergamotene	12.479	0.37	1658
Methyl eugenol	12.594	0.61	1650
β-Bisabolene	12.783	1.31	1637
Caryophyllene oxide	13.427	1.28	1615

**Table 3 microorganisms-13-02585-t003:** Results of the evaluation of temperature (Tsol–gel) and time (Tsol–gel) at 37 °C.

Groups	Temperature (Tsol–Gel) (°C)	Time (Tsol–Gel) 37 °C (s)
P	28 ± 1	39.31 ± 6.53
PLS0.2	26 ± 1	43.13 ± 3.72
PLS0.5	24 ± 1	26.52 ± 5.05
PLS1	20 ± 1	23.32 ± 3.63
PC	28 ± 1	46.46 ± 8.89
PCLS0.2	26 ± 1	38.62 ± 8.64
PCLS0.5	24 ± 1	27.48 ± 7.97
PCLS1	20 ± 1	18.76 ± 1.24

s: seconds.

**Table 4 microorganisms-13-02585-t004:** Evaluation of initial pH and after 30 days of storage at 25 ± 2 °C, 4 ± 2 °C, and 45 ± 2 °C.

Groups	Initial pH ± SD	pH Value After 30 Days (% of Variation *)		
		4 ± 2 °C	25 ± 2 °C	45 ± 2 °C
P	6.75 ± 0.03	6.87 (1.68)	7.11 (5.23)	7.15 (5.86)
PLS1	6.96 ± 0.03	6.96 (0.05)	6.97 (0.10)	7.12 (2.35)
PLS0.5	6.84 ± 0.01	6.93 (1.27)	6.80 (0.54)	7.05 (3.07)
PLS0.2	6.8 ± 0.01	6.89 (1.22)	7.07 (3.82)	7.2 (5.73)
PC	5.82 ± 0.03	5.85 (0.34)	5.67 (2.63)	5.98 (2.69)
PCLS1	5.74 ± 0.01	5.78 (0.75)	5.64 (3.54)	5.88 (2.38)
PCLS0.5	5.83 ± 0.02	5.85 (0.29)	5.95 (1.94)	5.95 (1.94)
PCLS0.2	5.88 ± 0.01	5.87 (0.23)	5.89 (0.23)	6.01 (2.15)

* % difference to initial pH.

## Data Availability

The original contributions presented in this study are included in the article. Further inquiries can be directed toward the corresponding author.

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
