# Peer review of "In Situ Gel Containing Lippia sidoides Cham. Essential Oil for Microbial Control in the Oral Cavity"

_microorganisms, 2025, doi:10.3390/microorganisms13112585_

Round 1

Reviewer 1 Report

Comments and Suggestions for Authors
  1. The manuscript is well-written but has minor grammatical errors
  2. Some sentences are overly complex
  3. The study design is robust, combining phytochemistry, material science, and microbiology. GC-MS, FTIR, rheology, and MIC assays are appropriately selected. But missing the details on in vivo safety (cytotoxicity) and release kinetics of LSEO from gels.
  4. The premise—using LSEO-loaded mucoadhesive gels for oral SSIs—is novel and justified. Poloxamer-chitosan synergy for thermogelling/mucoadhesion is well-supported by literature. Data support conclusions (e.g., thymol as major component, MIC reduction in gels). But Lack of statistical analysis for rheological/MIC data (e.g., p-values).
  5. CLSI guidelines are followed for MIC assays. No replicates mentioned for GC-MS or FTIR (only "triplicate" for DLS/zeta potential).
  6. Replace Microbiological Control with Microbial Control for precision in the Title
  7. Clarify "MIC values above 5 µg/mL" → "MIC values as low as 5 µg/mL."
  8. Add thermosensitive gel and oral mucoadhesion in keywords
  9. Cite recent reviews on EO-based antimicrobials (e.g., 2023/2024 papers).
  10. Justify chitosan concentration (0.024%) with a reference.
  11. Specify centrifugation speed in g (not just rpm).
  12. Clarify 3 cycles of refrigeration/agitation (time/temperature).
  13. Report retention indices (RI) for compound identification.
  14. Label peaks in Figure 1 (e.g., thymol C–H bend at 808 cm⁻¹.
  15. Explain high PDI (0.39±0.17) in PCLS0.5—is aggregation occurring?
  16. Add scale bars to Figure 3.
  17. Define "tanδ" in the caption of Figure 4b.
  18. Include turbidity or particle size changes over 30 days.
  19. State if resazurin was used for bacteria and fungi (methods imply only bacteria).
  20. Highlight why PCLS1 outperformed free LSEO (controlled release? Synergy?).
  21. Compare your MICs with other LSEO delivery systems (e.g., nanoemulsions).
  22. Propose future steps: in vivo efficacy, toxicity, clinical translation.
  23. Update pre-2020 references with recent studies (e.g., on poloxamer-chitosan gels).
  24. Figure 2: Add error bars to (a)–(c).
  25. Confirm SisGen registration covers in vitro use of LSEO.
  26. Line 44: "triclosan's" → "Triclosan’s" (capitalize trade names).

Final Recommendation

This manuscript is scientifically sound and innovative but requires minor revisions for clarity, statistical rigor, and alignment with recent literature. The gel system shows promise for oral antimicrobial applications, warranting further in vivo validation.

Reviewer 2 Report

Comments and Suggestions for Authors

Remarks
1. The choice of Lippia sidoides Cham oil as an antimicrobial agent compared to Rosmarinus officinális oil should be explained.
2. The Introduction section should contain references to modern publications devoted to the use of plant extracts in dentistry and wound healing (see work doi: 10.59761/RCR5108).
3. The water absorption capacity of the gels and the diameters of the zones of inhibition of microorganism growth in wells with Mueller-Hinton agar should be determined.
4. The shift of the 1590 to 1577 cm−1 vibration band in the IR spectrum should be shown and a diagram of the observed interactions during gel formation should be provided.
5. Photographs of the obtained gels and their behavior at physiological temperatures should be provided.
6. The Conclusions section should indicate the areas of application of the obtained gels.

Round 2

Reviewer 2 Report

Comments and Suggestions for Authors

The sentence “The band at 1577 cm−1 is the amine II 261 vibration of chitosan that is slightly shifted from 1590 to 1577 cm−1 indicating interactions 262 between the amide group from chitosan and O–H bond from poloxamer [28]” should be corrected. According to work [28], the amide group has a higher wave number, and the amine group has a lower one.

Author Response

R2

Comments and Suggestions for Authors

The sentence “The band at 1577 cm−1 is the amine II vibration of chitosan that is slightly shifted from 1590 to 1577 cm−1 indicating interactions between the amide group from chitosan and O–H bond from poloxamer [28]” should be corrected. According to work [28], the amide group has a higher wave number, and the amine group has a lower one.

R: We appreciate your insightful comment. Indeed, the text highlighted by Reviewer II is present in the manuscript, but it appeared in our first submission, which has already undergone revisions during Round 1. This excerpt was modified in Round 1 to the following version:

The chitosan amide II vibration band, commonly identified at 1577 cm−1 in the literature, was observed at 1577 cm-1, suggesting interactions between the amide group from chitosan and O–H bond from poloxamer.

In the current version, we clarified that the amide II band corresponds to the vibration observed at 1577 cm⁻¹, rather than the amine II band. Furthermore, in this new revised version (Round II), we took the opportunity to refine this excerpt as presented below:

The chitosan amide II vibration band, commonly identified at 1590 cm−1 in the literature, was observed at 1577 cm-1, suggesting interactions between the amide group from chitosan and O–H bond from poloxamer.